# Pitfalls in Oncogeriatrics

**DOI:** 10.3390/cancers15112910

**Published:** 2023-05-25

**Authors:** Silvio Monfardini, Francesco Perrone, Lodovico Balducci

**Affiliations:** 1History of European Oncology Program, European School of Oncology, 20121 Milan, Italy; 2Director Clinical Trial Unit, National Cancer Institute, IRCCS Fondazione G. Pascale, 80131 Naples, Italy; 3Oncology and Medicine, University of South Florida College of Medicine and Division of Geriatric Oncology, Senior Adult Oncology Program, H. Lee Moffitt Cancer Center & Research Institute, Tampa, FL 33620, USA

**Keywords:** cancer in the elderly, geriatric oncology, oncogeriatrics

## Abstract

**Simple Summary:**

The increasing age-related cancer incidence being a major challenge in healthcare, the best approach for the management of cancer in the elderly relies on the structured cooperation among clinical oncologists, geriatricians, and other specialists. The diffusion of this special approach is, however, quite limited in high-income countries and is almost absent in those with lower incomes. This commentary aims to shed light on the complex reasons of the somewhat surprising lack of spread of programs for cancer in the elderly. The description of the many factors in dealing with the implementation of such programs might be useful in suggesting possible solutions.

**Abstract:**

An oncogeriatric interdisciplinary activity exists only in a minority of high-income countries, and it is almost absent in those with lower incomes. Considering topics, sessions, and tracks in the main meetings and conferences of the major Oncological Societies in Europe and worldwide, the USA excluded, little attention has thus far been paid to the problem of cancer in the elderly. Again, with the exception of the USA, the major cooperative groups, for example, the EORTC in Europe, have only dedicated marginal attention to the research of cancer in the elderly. Despite major shortcomings, professionals interested in geriatric oncology have taken a number of important initiatives to highlight the benefits of this particular activity, including the organization of an international society (Société Internationale de Oncogeriatrie, or SIOG). In spite of these efforts, the authors believe that the management of cancer in the older population is still encountering several important and generalized pitfalls. The main obstacle is the grossly inadequate number of geriatricians and clinical oncologists necessary to an integrated care of the ever-expanding aging population, but other hurdles have been reported. Additionally, the prejudice of ageism can lead to missing potential resources for the development of a generalized oncogeriatric approach.

## 1. Introduction

The incidence and prevalence of malignant diseases are increasing with the aging of the population.

In response to the unrelenting epidemics of cancer in older aged persons, a number of clinical investigators have studied the management of cancer in the aged. This effort involved the cooperation of physicians, nurses, and allied health professionals focused on oncology or geriatrics. A number of conclusions were obtained and summarized in clinical guidelines issued by major professional societies [1,2,3,4,5].

All guidelines concord on the need to provide personalized care to older patients with cancer based on physiological rather than chronological ages and on individual social, economic, and emotional needs. The time-honored instrument to assess physiological age and individual needs, as well as individual expectations, is a comprehensive Geriatric Assessment. If a full assessment is excessively burdensome to some practices, the utilization of a shortened screening instrument is recommended. A corollary of these guidelines holds that the ideal situation would involve a special unit where the patients would benefit both from oncological and geriatric care [1]. Such units represent the rule in France where geriatricians and oncologists cooperate in all major hospitals and are present in some major institutions of the USA, Canada, Europa, and Australia [1,6,7].

When the ideal solution is not obtainable, alternative solutions were proposed, such as including a geriatric section in the curriculum of oncology and hematology training or to endorse training programs providing a double certification both in oncology and geriatrics [8,9].

In this commentary, we will examine which obstacles may prevent the implementation of the initiatives taken worldwide to improve the care of older patients with cancer.

## 2. The Landscape

The geriatric assessment has been implemented in 30–50% of the oncology practices in the USA, despite its well documented benefits on treatment outcome that include the prolongation of active life expectancy and improved patient satisfaction [1,6,8,9,10].

Clearly, these results are dismal, as the geriatric assessment, especially in its abbreviated forms, represents a simple and relatively inexpensive life-saving practice.

The French experience instructs us that a routine use of geriatric assessment in the management of older patients with cancer may be obtained only when a structured cooperation between oncologists and geriatricians exists. One may assume that the implementation of the geriatric assessment is even scarcer in other countries due to the shortage of specialists.

Except for France, where they represent the rule, geriatric oncology clinics are found only in a minority of countries and even there only in few institutions [7]. The attempts to institute these clinics have been limited to high-income countries, but even some of these countries still lack the resources to enroll geriatricians and other professionals necessary to staff a geriatric team. The complexity of the composition of these geriatric oncology programs explains why an extra supply of time and personnel is needed. In fact, the team of the health professionals should be composed by medical oncologists, surgical oncologists, radiotherapists, and geriatricians, along with a dietician/nutritionist, social worker, nurse, physiotherapist, pharmacist and, of course, primary care physicians.

While the American Society of Clinical Oncology has supported the educational and research efforts of geriatric oncologists, the major oncological societies in Europe and worldwide have by and large sidelined or ignored the issues related to the aged. The major cancer cooperative groups around the world have dedicated only marginal attention to the aged, with the exception of the American group Alliance (former CALGB) that has supported the validation of the geriatric assessment in clinical research [11], has promoted randomized clinical trials in older patients [12], and has collected an important database to study metabolic markers of aging [13]. Some of the authors of these studies were geriatricians, as the Alliance implemented the teamwork of oncologists and geriatricians at the cooperative research level. Other cooperative groups in Europe have occasionally conducted important studies dedicated to older patients [14,15,16,17,18,19,20].

In the USA, combined specialty training in geriatrics and oncology has been implemented [8]. Some of the individuals who had received combined training are among the most prominent investigators and educators in the field, but the number of trainees has been inadequate for fulfilling the workforce demand. To our knowledge, this approach has not been adopted by any other training program [9].

Despite major shortcomings, professionals interested in geriatric oncology have taken a number of important initiatives to highlight the benefits of this particular specialty. They include:the organization of an international society (Société Internationale de Oncogeriatrie, or SIOG), with hundreds of members from throughout the world [3]. The society meets yearly and has been issuing the Journal of Geriatric Oncology since fourteen years.The formation of cooperative research groups dedicated to clinical research in geriatric oncology. The most successful model of these groups that has spawned around the word is the Cancer and Aging Research Group (CARG) that was founded by the late Arti Hurria, counts on members from throughout the world, and includes geriatricians, oncologists, nurses, and other professionals involved in elderly care [21].

## 3. Obstacles

### 3.1. Geriatrics Specialists

By and large, the number of geriatric specialists necessary to take care of the ever-expanding aging population is grossly inadequate. This is true not only worldwide but even in developed high-income countries [22].

Geriatric medicine is a recognized independent medical specialty in 17 of 31 European countries, a recognized subspecialty of internal medicine in 10 countries, and, in two countries, is endorsed as both an independent specialty and a subspecialty of Internal Medicine [23]. There is considerable variability in the length and content of professional training [24,25]. In Europe, the attendance of Postgraduate training in geriatrics is often heterogeneous, and there is a wide variation in the number and distribution of geriatricians [24,25,26,27]. In the USA, Postgraduate training in geriatric medicine is available and structured with a similar curriculum for the whole country. However, the demand for training is lower than the positions offered. As a consequence, the supply of geriatricians becomes progressively more inadequate for the ever increasing need [28]. In Canada and in Australia, geriatric medicine is also recognized as a specialty [29]. In Latin America, the development of geriatrics is heterogeneous, and it reflects the economic disparities of that continent. In general, geriatric medicine is recognized as a medical specialty in the countries with higher incomes. 

Geriatric Oncology is early in Africa and still not existent from a practical stand point. The older people comprise in fact only a small part of the population [30].

This worldwide paucity begs two questions: what is the role of the geriatrician? Why is the geriatric specialty considered one of the least desirable by young physicians?

Clearly, the geriatrician cannot be expected to act as a primary care physician of all individuals aged 65 and over without special aging concerns. When available, the geriatrician may become the primary care provider to patients with a geriatric syndrome and a consultant for patients undergoing a dangerous procedure such as hip replacement or cancer chemotherapy. One may argue that a geriatrician should provide primary care to the patients considered frail (1–5) and to those over 85 or 90. However, even when one limits the scope of geriatrics, the expertise available is inadequate to care for the emerging needs, mainly, but not only, in low-income countries. Additionally, the disproportion between providers and needs may keep increasing, as the population over 85 is experiencing the most rapid growth.

Several solutions have been proposed to solve this problem. It is generally accepted that the curriculum of medical schools and of different medical and surgical specialties should include geriatric modules, enabling future practitioners and specialists, at the very least, to recognize a geriatric emergency, such as delirium, and to decide when a geriatric consultation is indicated. The implementation of this recommendation is still developing even in the Western world, especially for what concerns Undergraduate education [27].

The difficulties in recruitment into geriatrics have multiple causes. Of these, there are two that are prominent: the complexities of the older patient and economical remuneration. The presence of multiple chronic conditions frustrates the practitioner’s aspiration to provide effective and quick-working solutions and compels the practitioner to walk new uncharted and dangerous clinical roads. In most countries, physicians are reimbursed for the quantity and not the complexity of care. In the hour that it takes to evaluate the mental status, medication list, and social support of an 80-year-old patient, a primary care physician may see six younger patients with mild respiratory infections, be paid six times as much, and have lower risk of malpractice. The economic disproportion is even more accentuated when the income of geriatricians is compared with that of practitioners performing procedures that currently are remunerated at a much higher level.

Unconscious ageism may play a part as well. The maintenance of the health of a person with multi-morbidity, disability, and/or a short life-expectancy, whose condition is mostly irreversible and whose management is costly in economic and human terms, may feel futile in the investment of time and resources. Older individuals, especially those who are disabled, are resented for living too long and for interfering with the growth of the young ones. Not too long ago, the Governor of Colorado Richard D Lamm asserted that “Older people have a duty to die”. In 2014, a prominent oncologist and medical ethicist wrote an article for the Atlantic Magazine entitled “Why I hope to die at 75” [31]. In this article, Dr. Emanuel proclaimed that most curative treatments are futile for people of an advanced age. Given the professional stature of this world-renowned author, it is not surprising that young physicians perceive his “boutade” as scorn for geriatric practice and geriatricians.

The scarcity of geriatricians represents a major impediment to the institution of any form of gero-oncology cooperation. Furthermore, the number of geriatricians available in the outpatient setting is limited, as most of these practitioners are involved in assisted living facilities, SNIFF (Skilled Nurses Facilities) units, palliative care units, or are specialized in specific geriatric syndromes such as dementia or falls. Last but not least, many geriatricians are suspicious of aggressive cancer treatment in elderly patients [32,33], which sometimes is compared to elderly abuse.

In several institutions, there might indeed be geriatricians, but their availability for the interdisciplinary approach with clinical oncologists may be hampered by the fact that they may be involved with other multiple roles and suffer from a time constraint due to the overwhelming number of elderly cancer patients. Then, if geriatricians are requested for an interaction, may feel confined to the periphery of the organization of cancer treatment [33]. The team-work allows geriatricians to witness the benefits of chemotherapy, even in some frail individuals, and to establish trust from the oncologists who are willing to heed the geriatricians’ concerns.

### 3.2. Oncology Specialists

In 2007, the American Society of Clinical Oncology Workforce Study lamented a scarcity of oncology specialists due to two factors: increased incidence of cancer due to the aging of the population; and the increased prevalence of cancer due to improved treatment and prolonged survival of cancer patients [34]. The study predicted a critical shortage by 2020. A recent update of the study confirmed this prediction [35]. In addition to the increased incidence and prevalence of cancer, other factors contribute to the scarcity of oncologists. They include maldistribution between rural and urban areas, increased retirements of aging specialists, physician burn-out, increased complexities of cancer treatment that compel specialists to focus on the management of few diseases. This being the situation in the USA and high-income countries, one may expect an even more critical shortage in low-income countries.

The modern practice of oncology is based on team-work expressed in the tumor boards, where different experts are consulted to generate a treatment plan. This familiarity with team-work does not necessarily mean that oncologists are ready to work with geriatricians. The oncological and the geriatric culture may be quite apart [33,36].

Though dealing with palliative medicine and end-of-life care might have sensitized oncologists more than other medical specialists to individual patient needs and expectations, by and large, oncologists are trained to embrace evidence-based medicine based on the results of randomized clinical trials. This attitude has been reinforced by the rapid development of precision medicine [37]. This cross-seeding of information opinion and expertise should happen, ideally, in a geriatric oncology tumor board, but the institution of such a board is complex for several reasons, including the involvement of different oncological expertise. Additionally, the availability of geriatricians, at present and in the predictable future, is inadequate to supply different tumor boards focused on specific neoplasms. At the very least, it is compelling that oncologists have become familiar with specific geriatric concepts, including impairment, disability, frailty, active life expectancy, and compression of morbidity. In turn, geriatricians should learn what is the significance of a complete and partial response to treatment, disease, and event-free survival.

The SIOG and ESMO/ASCO recommendations for a Global Curriculum in medical oncology have promoted geriatric knowledge and skill for future medical oncologists [38]. However, with the exception of the institutions where a clinical activity of geriatric oncology is already ongoing, the information has been thus far not incorporated in the Postgraduate medical oncology curriculum in the universities of Europe, South East Asia, and Australia.

Lack of information is also a problem. Traditional clinical trials have very restrictive and exacting recruitment criteria that make the enrollment of older individuals particularly challenging [39,40,41]. Inadequate access to transportation and social support may further hamper the conduction of clinical trials in older patients. Many oncologists are aware of the benefits of the geriatric assessment, but they are unable to implement this instrument due to scarcity of time and limited resources, particularly in low-income countries [1,32,42]. The Association of Community Cancer Centers (ACCC) has identified three barriers to the implementation of a geriatric assessment in patients with cancer: limited time, insufficient personnel, and inadequate information [1].

There is, however, a deeper reason, which may be related to unconscious ageism. The main focus of the training in medical oncology—as it is in other medical and surgical branches—is to provide a rapid solution of the medical problems rather than getting involved in the complex, time consuming, and often frustrating care of an older person. The most important result of this care may be represented by the appreciation of the patient rather than the disappearance of the cancer and the prolongation of survival. Additionally, one should add the scarce support of a culture that arguably considers the older and the disabled a burden to eliminate.

### 3.3. Ageism

Kahli E. Zietlow and Serena P. Wong have been writing that there is the diffusion of the prejudice of ageism within society [43]. People who hold ageist views believe that all people of old age are of declining intelligence, unable to change or learn, rigid, and dull. They assume that any physical or mental change is due to the ageing process and is, therefore, untreatable. Ageist beliefs at the society level can lead to missing the potential contributions of older persons, therefore, providing inadequate societal resources for the care of older patients who may be considered disposable.

Though this prejudice has been already disproved 2500 years ago in Ancient Greece, it is still alive and thriving today. During the golden age of Athens, the children of the tragic writer Sophocles tried to gain control of his assets by declaring him incapacitated, as he was over 90. After Sophocles read excerpts from his last work to the judges, Oedipus in Colonus, they condemned his children for false pretense.

This article has already reviewed how ageism may influence the scarcity of geriatricians and may discourage oncologists from participating in the clinical trials of older patients. Perhaps, the most pervasive and yet concealed attitude toward older individuals is resentment, resentment that is shared even by the young old toward the oldest old and by the well-to-do old versus the poor old.

## 4. Conclusions

The results of this analysis are summarized in Table 1 and Table 2.

This brief commentary highlighted that:The optimal treatment of older persons with cancer may involve a personalized plan of care obtained by the close cooperation of oncology and geriatrics professionals. This cooperation should extend to clinical research involving the aged.This cooperation has been implemented only occasionally in high-income countries and is all but absent in middle- and low-income countries.The implementation of this system is embattled by many factors that include the scarcity of manpower among the most important factors.The prejudice toward older people both at the level of professionals and society exerts a negative influence in the generalized spread of the oncogeriatric approach.An additional problem that is not acceptable to mention in mixed company, but certainly underlies the resentment of older and disabled individuals, is the burden these individuals represent to the society. This includes the burden of financial resources, the burden on health care facilities, and the burden on the caregiver. Nobody has a definite response to this bulging problem, but the solution cannot consist of denying care to the patients based on chronologic age or disability, nor can it consist of ignoring the issue. We believe that the cooperation of oncologists and geriatrics specialists may represent a necessary initial step to solve this issue. The cooperation may allow the team to present the definitive benefits, risks of each treatment approach, and allow the patients to choose the course of action most fitting the individual condition.

## Figures and Tables

**Table 1 cancers-15-02910-t001:** clinical initiatives.

Special geriatric oncology units	Implemented throughout France Limited to major institutions in USA, Canada, Europe, and Australia
Professional education through major professional association	ASCO has a curriculum in geriatric oncology and reserves a section of the annual meeting to the issue ASCO has a special interest group in cancer and aging AGS has a special interest group in cancer and aging AACR is organizing a committee in cancer and aging
Geriatric module as part of the training in hematology oncology	USA only
Double clinical training in oncology and geriatrics	USA only
Dedicated clinical trials in the elderly in major oncology group	Alliance group (former CALGB) in the USA Occasional studies dedicated to older patients in Europe

**Table 2 cancers-15-02910-t002:** Obstacles.

Manpower scarcity and distribution
Professional interest in aging
Ageism
Geriatrician’s tools and know-how, often perceived ambiguously
Oncologists unclear as to the Geriatricians role
Lack of Governmental initiatives to promote studies of cancer in the elderly

## Data Availability

The data can be shared up on request.

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
