# Peer review of "Pitfalls in Oncogeriatrics"

_cancers, 2023, doi:10.3390/cancers15112910_

Round 1

Reviewer 1 Report

This is an excellent opinion paper that comes in due time. The ideas raised are meaningful and well argumented. It is based on precise observations and studies. The comparison of the situation between countries is sound and useful.

The addition of a final part, outlining clues for decision makers, learned societies, colleagues and the society, would be a plus. This can be summarized in a second table. The paper already outlines many ideas such as early education about geriatry during oncology board course, initiatives of learned societies such as ASCO or decisions of countries such as France, which may be adapted by the others. Further ideas may also be proposed about biological research, phase III trials and new research methods, role of advance practice nurses or fight against ageism.

None. Few edits along the manuscript.

Author Response

To answer Your question I would add at the end of the Conclusions:

The research activity can be definitely useful in providing a further impulse to the spreading of the Oncogeriatrics.Also in this field new avenues can be opened by the genomic revolution.However differently from the past ,phase 2 and 3 trials should be designed and conducted taking into account the age associated variables.For this purpose the geriatric partecipation is essencial.One simple final clue to improve the clinical practice for decision makers,learned societies,collegues and society could essencially consist, in presence of an old cancer patient ,of a recommendation to perform at least a short geriatric screening.Even with the help of an advanced practice nurse

Reviewer 2 Report

This is a very interesting commentary with many insightful elements on this important topic. The introduction and landscape sections are very well documented and highly informative. The sections on geriatrics and oncology specialists give interesting clues on potential obstacles. Elements regarding the lack of collaboration and its  deteminants could probably be more discussed, as well as author's thoughts on why such collaboration could be implemented in France but not in other high income countries. However, the last section on ageing and putative society resentment towards older people appears poorer and a bit redundant. I also did not found the example on Florida very convincing and in my opinion it weakens the conclusion. Relatedly, the [44] reference does not feel relevant with the text calling that reference.    

Author Response

Concerning a further discussion on the lack of collaboration between Oncologists and Geriatricians, a phrase has been added to underline that a true methodologic approach  on when an how this interplay should take place has never been so far developed.

To decrease the redundacy the example on Florida has been eliminated

Reference 44 has been eliminated ,as requested